# Claustrophilic oviposition: Oviposition performance depends on container size in a novel forced oviposition method for *Culex quinquefasciatus* and *Aedes aegypti*

Kendra A. Dagg[1], Alden S. Estep[2], Cason E. Bartz[3], Edwin R. Burgess I.V.[1]*

**1** Entomology and Nematology Department, Institute of Food and Agricultural Sciences, University of Florida, Gainesville, Florida, United States of America, **2** Mosquito & Fly Research Unit, USDA ARS Center for Medical, Agricultural, and Veterinary Entomology, Gainesville, Florida, United States of America, **3** Gainesville Mosquito Control Services, Gainesville, Florida, United States of America

* edwinburgess@ufl.edu

## Abstract

Mosquito vector control research relies heavily on the use of live organisms, with wild populations playing a critical role in surveillance, field product validation, and strengthening the overall efficacy and accuracy of studies. However, too often wild mosquito rearing and colony establishment in laboratory settings is consistently challenging. Here we present a modified version of a forced oviposition method for use against two different mosquito species, *Aedes aegypti* and *Culex quinquefasciatus*. The efficacy of the technique was tested with both laboratory and wild-caught strains placed in oviposition tubes consisting of a 1.5 mL tube containing moistened cotton and a strip of germination paper. To determine if size had an impact on oviposition rate, an additional test was conducted using four different size oviposition tubes; 1.5, 5, 15, and 50 mL. Overall, the forced oviposition method was highly effective, successfully generating eggs from lab reared and field collected strains. The method was more effective with *Ae. aegypti* (>80% females oviposited) compared to *Cx. quinquefasciatus* (50–60%) in both strains. *Culex quinquefasciatus* holding time was longer (3–7 days) compared to *Ae. aegypti* females, which oviposited within 24 hrs of being transferred into tubes. Intact versus broken egg rafts affected egg hatch rate in laboratory *Cx. quinquefasciatus* strains but had less impact on the wild strain. Additionally, both *Ae. aegypti* and *Cx. quinquefasciatus* displayed a claustrophilic oviposition behavior with a higher percentage of females ovipositing in the 1.5 and 5 mL tubes. This study demonstrates that the forced oviposition method can be easily adapted to other mosquito vector species and effective in producing $F_1$ progeny needed for critical vector research.

**Data availability statement:** Data is available at the Institutional Repository @ University of Florida (IR@UF) https://ufdc.ufl.edu/ir00012296/00001/downloads.

**Funding:** This project was funded and supported, in part, by the Florida Department of Agriculture and Consumer Services, Division of Agricultural Environmental Services (Project #31236 to ERB), as well as the US Department of Defense Deployed Warfighter Program (Project #W911QY2010003 P00001 to ERB). The funders had no role in study design, data collection and analysis, decision to publish, or preparation of the manuscript.

**Competing interests:** The authors have declared that no competing interests exist.

## Author summary

Mosquito vector control programs rely heavily on data and information generated from research and phenotypic insecticide resistance monitoring, which require the use of live mosquitoes, both from laboratory colonies and field caught populations. Here we tested a modified version of a force oviposition technique originally designed for establishing wild *Anopheles spp.* colonies, on two key mosquito vectors, *Culex quinqufasciatus* and *Aedes aegypti*. We find that, despite their distinct oviposition preferences and behaviors, both species of mosquitoes generated adequate numbers of $F_1$ eggs needed to start either laboratory or wild colonies. A key advantage to this method is its ease of use, only requiring minimal numbers of female mosquitoes and inexpensive, easily accessible materials commonly found in most laboratories and mosquito control districts. Further application of the forced oviposition method has important implications for enhancing our capacity to conduct a range of vector research with various key mosquito species. This could include key vector research projects such as, determining single female sex ratios, producing genetic isolines by controlling specific genetic backgrounds, and conducting heritability studies focused on selective mating.

## Introduction

Vector-borne disease (VBD), and particularly mosquito-borne disease, such as malaria, dengue virus, and West Nile virus, are major causes of morbidity and mortality [1]. In 2023, the World Health Organization (WHO) estimated that VBDs accounted for 17% of all reported global infectious disease cases [2]. Despite continual efforts to reduce this global disease burden, an increase in disease incidence is thought to be due to a combination of factors, such as shifts in climate, land-use practices, international travel and trade, and urbanization, all of which affect both the pathogen and their associated vector [1]. The number of reported malaria cases in 2023 surpassed that of 2022 by nearly 11 million cases [3]. By April 2024, over 7.6 million dengue cases were reported from roughly 86 countries/territories, which was three times the number reported in 2023 [4]. While substantial medical advances (e.g. vaccine development) have contributed to case management, costs of medical care, supply limitations, and constraints on eligibility for medical treatment (e.g. age or medical status) restrict access to care [5]. To date, vector control remains the principal method of any vector-borne disease control program, and in some cases, vector control is currently the only method available for disease prevention [6,7].

The development and testing of mosquito vector control tools and strategies relies heavily on research conducted on the target mosquito species itself. The use of live organisms is vital for research on mosquito behavior, biology, physiology, product testing, insecticide resistance, vector incrimination, and vector competency [7–9]. Additionally, wild collected populations play a pivotal role by providing insight into the field applicability of methods and strengthening the overall efficacy and accuracy of

studies. In order to properly conduct mosquito research, large numbers of individual organisms are required to ensure the robustness of the dataset. While most strains of mosquitoes are hardy and resilient in their natural habitat, wild mosquito rearing and colony establishment in laboratory settings is consistently challenging [8]. Quite often, the most common barriers to insectary rearing of wild mosquitoes are insufficient numbers of females collected from the field, and a general difficulty to induce oviposition from field blood-fed or gravid females [9].

To address these rearing challenges, we sought to modify an oviposition encouragement technique originally developed by Morgan et al. [10] for establishing laboratory colonies of field-caught *Anopheles spp.*, for use with two different mosquito species of public health importance, *Culex quinquefasciatus* and *Aedes aegypti*. Both vector species have distinct oviposition habitat preferences and oviposition behaviors. The Morgan et. al [10] method has been successfully used to establish wild laboratory colonies for multiple *Anopheles spp.* in Thailand, Ethiopia, Madagascar, and Uganda [9–12]. However, there is no literature on similar techniques available used to establish colonies of other mosquito vector species. Here we demonstrated that with slight modifications, this method was effective at producing high numbers of viable *Cx. quinquefasciatus* and *Ae. aegypti* eggs from both established laboratory strains as well as $F_1$ progeny from field-caught female mosquitoes. Additionally, Morgan et al. (2010) [10] hypothesized that the success of the method could be related to the confined space the mosquitoes are put in, which may suggest an affinity for tight spaces (i.e., claustrophilic) when ovipositing. To test this hypothesis, we investigated if holding container size had an impact on oviposition rate and egg production in both species through the use of four different sized oviposition containers that are commonly available in laboratories and mosquito control districts.

## Materials and methods

### Mosquito strains and rearing

Laboratory reared strains and locally collected wild populations of *Cx. quinquefasciatus* and *Ae. aegypti* were used in this study. All laboratory strains were provided by the Center for Medical, Agricultural & Veterinary Entomology (CMAVE), U.S. Department of Agriculture (Gainesville, FL, USA). Laboratory strains used for this study were the *Cx. quinquefasciatus* CMAVE strain and the *Ae. aegypti* 'Orlando' (ORL) strain, each referred to hereafter as the "lab strain" for their respective species. Laboratory strains were reared using standardized protocols previously described [13]. Wild caught mosquitoes were supplied as adult blood engorged or gravid females for both species by Gainesville Public Works Department, Mosquito Control (Gainesville, FL, USA). Mosquitoes were collected by the city Mosquito Control Department in July 2024 within the Gainesville, FL city area and were morphologically identified using keys from Darsie et al. [14]. Gravid females were held for 24 hrs under standard insectary conditions (27°C, 70% relative humidity, light:dark cycle 12:12 hrs), and blood engorged females were held for 96 hrs before use in any experimentation.

### Experiment 1: Forced oviposition performance with recently collected field strains

Eggs were generated using a modified version of the forced-oviposition method originally designed for field caught *Anopheles spp.* described by Morgan et al. [10]. Modifications to the *Anopheles spp.* method were based on genus oviposition behavior and ease of equipment accessibility in an insectary. Each oviposition container consisted of a 1.5 mL Eppendorf tube, moistened cotton, and a strip of germination paper (Fig 1A). A small hole was made in the cap of the tube to allow air flow during the holding period. Cotton balls were broken into smaller pieces and were moistened using Gainesville city tap water. At the bottom of the 1.5 mL tube, a single small piece of moistened cotton was pushed down into the conical tip of the tube. Once pushed down, the cotton filled no more than the conical tip of the 1.5 mL tube and all excess water was removed. A roughly 2 x 1 cm strip of germination paper, with a tapered end (forming a triangle like shape) was inserted tip first into the tube. The inserted end of the germination paper was pushed down to fold over the surface of the moisten cotton (Fig 1B). For both laboratory and wild *Ae. aegypti*, excess water was completely removed from the bottom of the tube,

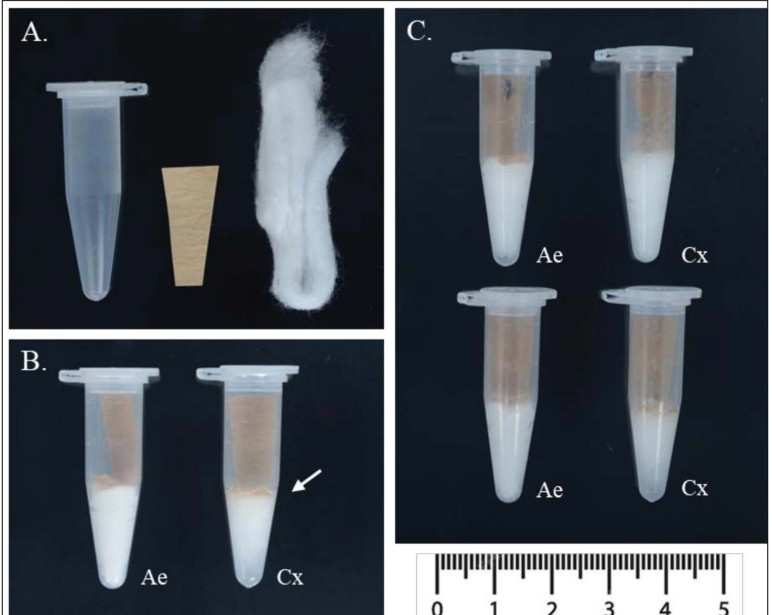

**Fig 1. Layout of equipment and examples of complete set up of modified forced oviposition technique for *Aedes aegypti* (*Ae*) and *Culex quinquefasciatus* (*Cx*).** (A) Oviposition containers consist of a 1.5 mL Eppendorf tube, a small piece of germination paper cut with a small taper at the end, and a piece of cotton. (B) Moistened cotton is pushed to the bottom of the tube and germination paper placed on top with only the tapered end covering the cotton. Excess water is removed from *Ae* tubes, with a ~1 mm water layer remaining (white arrow) only for the *Cx* tube set up. (C) A single female is aspirated into each container (Photo credit: Kendra A. Dagg).

however, for *Cx. quinquefasciatus* a thin layer of water was left at the bottom of the tube (approx. 1mm at the meniscus) (Fig 1B-C). Individual females that had previously been blood fed were gently aspirated into a prepared 1.5 mL Eppendorf tube and the cap was secured. Oviposition tubes were maintained in standard insectary conditions (27°C, 70% relative humidity, light:dark cycle 12:12 hrs) for the designated holding period. *Aedes aegypti* strains were checked at 24 hrs and 48 hrs, while the *Cx quinquefasciatus* strains were checked every 24 hrs for 168 hrs (7 days).

Variables collected included: 1) did the female oviposit (yes/no) (Fig 2A-B), 2) number of eggs laid, and 3) number of days between transfer to oviposition tube to laid eggs. Oviposition was determined through visual examination of germination papers and cotton under a dissecting microscope (Stemi 2000, Zeiss International, Germany). If females did not oviposit within the time period, they were dissected to check for gravidity (yes/no), i.e., to determine if they had eggs but did not oviposit. To check the viability of eggs, 20–25 randomly selected eggs from each female were transferred to a 35 mm diameter plastic Petri dish. If a female laid <20 eggs, all eggs available were hatched. Each dish was filled with 5 mL of tap water and 20 µL of a larval food slurry (approx. 150 mg of ground fish food flakes, mixed with 1.5 mL of tap water). Dishes were held in standard insectary conditions and checked every 24 hrs for 72 hrs. The total number of eggs hatched was recorded for each dish at each timepoint. Any hatched larvae were reared up to the L2 stage for ease of visualization.

For *Cx. quinquefasciatus*, the lab strain was replicated 13–18 times across each of four generational cohorts (N = 60) and the wild strain was replicated 11–13 times across two generational cohorts (N = 24). For *Ae. aegypti*, the lab strain was replicated 9 – 13 times across three generational cohorts (N = 33) and the wild strain was replicated 10 – 13 times across two generational cohorts (N = 23). This replication structure represents all individual females that met three conditions (in order): 1) blood fed, 2) alive at the end of the assay, 3) gravid. Any females that did not meet all three conditions were excluded from the analysis.

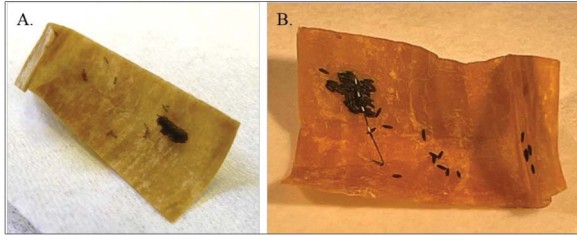

**Fig 2. Example egg papers collected from oviposition tubes.** Eggs were collected between 24 – 168 hr after female transferred into oviposition tube for *Culex quinquefasciatus* (A) and 24 hr for *Aedes aegypti* (B) (Photo credit: Kendra A. Dagg).

## Experiment 2: Tube size effects on forced oviposition performance

To determine if oviposition tube size influenced oviposition rate and egg numbers, oviposition tubes were prepared in four different sizes: 1.5 mL Eppendorf tubes, 5 mL polypropylene centrifuge tube, 15 mL polypropylene centrifuge tube, and 50 mL polypropylene centrifuge tube. The additional tube sizes were set up using the same equipment as previously described and capped with either a piece of cotton (5 mL tubes) or a piece of mesh secured by a rubber band (15 mL and 50 mL tubes) (Fig 3). Moistened cotton pieces and germination paper size was prepared proportional to the oviposition tube size. Gravid lab strain females were individually aspirated into an oviposition tube and stored in standard insectary conditions (27°C, 70% relative humidity, light:dark cycle 12:12 hrs). Oviposition was observed for the *Ae. aegypti* lab strain and the *Cx. quinquefasciatus* lab strain every 24 hrs for 48 and 96 hrs, respectively. The same data variables were recorded as described in the forced oviposition assay (Experiment 1).

For *Cx. quinquefasciatus*, all four tube sizes were replicated 8 – 10 times across each of three generational cohorts (N = 114). For *Ae. aegypti*, all four tube sizes were replicated 15 times across two generational cohorts (N = 120). This replication structure represents the same criteria for inclusion in the analysis as described in Experiment 1.

## Statistical analyses

All analyses were conducted in R version 4.4.2 (R Core Team 2024) and statistical significance was set to $\alpha = 0.05$. For Experiment 1, analysis was done in the following order for *Cx. quinquefasciatus*: 1) a generalized linear model (quasipoisson) was used to assess if there was a difference in the time to oviposit (dependent variable) between the lab and wild strains (independent variable). 2) A 2x2 Chi-square test of independence with Yates correction for continuity was done to assess differences in frequency of oviposition between the two strains. 3) To determine if there was a strain effect (independent variable) on the number of eggs laid by females (dependent variable), a general linear model was used. 4) To determine if there was a difference in hatch rate (dependent variable) by strain (independent variable), a generalized linear model (quasibinomial) was used. 5) Finally, the oneway.test function, which controls for heterogeneity of variance, was used to determine if there was an effect of egg rafts being broken up vs. intact (independent variable) on the hatch rate (dependent variable). Because this test is a one-way ANOVA, strains had to be analyzed separately. For Experiment 1, analysis of *Ae. aegypti* was identical to the *Cx. quinquefasciatus* analysis except that all *Ae. aegypti* females oviposited within 1 day, and they lay eggs singly rather than in rafts, thus only analyses 2 – 4 were done on *Ae. aegypti*. For Experiment 2, all analyses were identical to Experiment 1 except that the independent variable of 'strain' was replaced with 'tube size' and Chi-squares used Yates correction for continuity (unnecessary for 4x2 tables). All models were diagnosed for departures from statistical assumptions, included Q-Q plots of residuals, and tests for homogeneity of variances. When Chi-square expected cell means were less than 5, Fisher's exact test was used instead [15]. Where applicable, in the event the omnibus Chi-square test of independence or Fisher's exact test was positive, pairwise comparisons were done using Fisher's exact test. For generalized and general linear models, if overall significance was observed, and where applicable, posthoc pairwise comparisons were done with the 'emmeans' package [16].

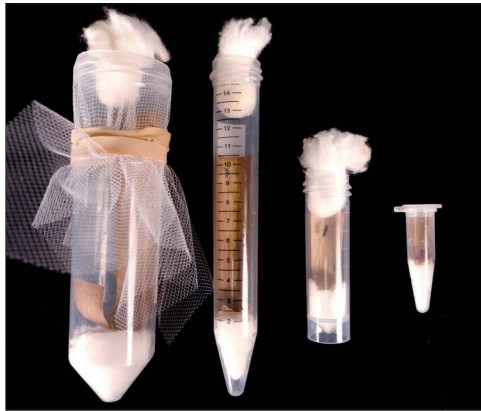

**Fig 3. Impact of tube size on oviposition performance.** Four different tube sizes were used to determine if tube size affected oviposition performance; (from left to right) 50 mL, 15 mL, 5 mL, and 1.5 mL (Photo credit: Kendra A. Dagg).

## Results

### Experiment 1: Forced oviposition performance with recently collected field strains

For *Cx. quinquefasciatus*, there was a significant difference between the strains in the time it took for females to oviposit (Fig 4; $F = 11.68$, df = 1, 446, $P = 0.001$), but whether females oviposited at all was independent of strain (Fig 5A; $P = 0.629$). When females did oviposit, there was a significant difference between the strains in the mean number of eggs oviposited (Fig 5B; $F = 20.72$, df = 1, 46, $P < 0.001$). There was no difference between the two strains in the number of eggs that hatched (Table 1; $F = 0.37$, df = 1, 46, $P = 0.544$). For the lab strain, there was a significant difference in the percentage of eggs hatching between intact and broken up egg rafts (Fig 6A; $F = 28.73$, df = 1, 13.48, $P < 0.001$), while no difference was observed in the wild strain (Fig 6B; $F = 2.82$, df = 1, 12.79, $P = 0.117$).

For the lab and wild *Ae. aegypti*, all females oviposited within 1 day of being added to the tube (Fig 3). Whether females laid eggs at all was independent of strain (Fig 7A; $P = 0.392$). When oviposition occurred, there was a significant difference in the mean number of eggs oviposited between the lab and wild strains (Fig 7B; $F = 47.48$, df = 1, 49, $P < 0.001$). There was no difference between strains in the number of eggs that hatched (Table 1; $F = 0.14$, df = 1, 49, $P = 0.707$). All females, regardless of strain, oviposited within 1 day of being added to the tube.

### Experiment 2: Tube size effects on forced oviposition performance

For lab strain *Cx. quinquefasciatus* there was no difference in the number of days until oviposition occurred among the tube sizes (Fig 8; $F = 0.25$, df = 3, 61, $P = 0.864$). The lab strain *Cx. quinquefasciatus* took $1.2 \pm 0.07$ (mean ± SEM) days to oviposit, regardless of tube size. The frequency of females that oviposited was dependent on the container size (Fig 9A; $\chi^2 = 11.2$, df = 3, $P = 0.011$). There were pairwise significant differences where oviposition was dependent on tube size, including between 1.5-mL and 15-mL tubes ($P = 0.004$), and 5-mL and 15-mL tubes ($P = 0.017$). No other pairwise comparisons were significant. There was no significant difference of the mean number of eggs oviposited among the tube sizes when females did oviposit (Fig 9B; $F = 0.20$, df = 3, 61, $P = 0.893$), nor was there a significant difference among the tube sizes in the number of eggs that hatched (Table 2; $F = 0.68$, df = 3, 61, $P = 0.565$).

However, there was a significant difference in the number of eggs that hatched when egg rafts were oviposited broken up rather than intact (Fig 10; $F = 5.73$, df = 1, 32.74, $P = 0.023$). Broken up egg rafts were generally in small clusters or singly laid eggs. Of the females that oviposited, 50 had intact egg rafts, with 15 females producing broken up egg rafts.

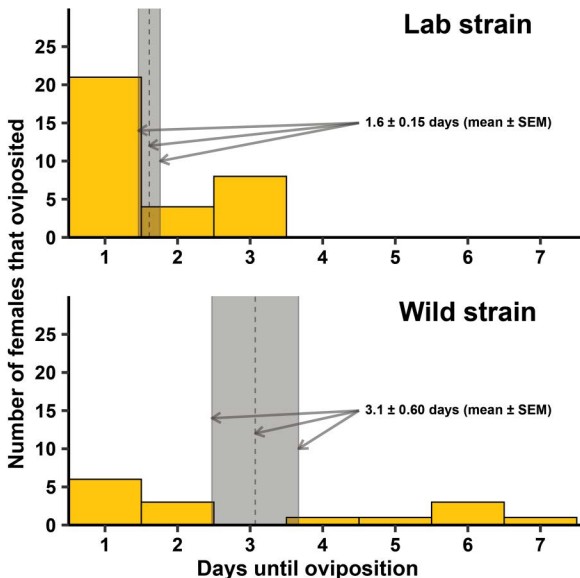

**Fig 4. Number of female lab strain *Cx. quinquefasciatus* that oviposited between 1 and 7 days in Experiment 1, utilizing a lab strain and a wild strain.** There was a statistically significant difference between the strains at α = 0.05, so mean ± standard error of the mean (SEM) was calculated for each strain separately. The mean days to oviposition was 1.6 ± 0.15 for the lab strain and 3.1 ± 0.60 for the wild strain (gray columns in the figures represent SEMs, with the dashed lines in the middle representing the means). The lab strain was replicated 13-18 times across each of four generational cohorts (N = 60) and the wild strain was replicated 11-13 times across two generational cohorts (N = 24).

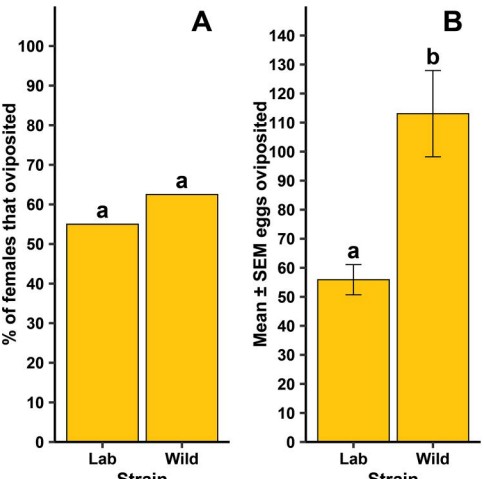

**Fig 5. Differences for forced oviposition success frequency (A) and mean± standard error of the mean (SEM) eggs when lab and wild strain female *Cx. quinquefasciatus* oviposited (B).** Different lowercase letters above bars indicate statistical significance of pairwise comparisons between strains, with significance set at α = 0.05. The lab strain was replicated 13-18 times across each of four generational cohorts (N = 60) and the wild strain was replicated 11-13 times across each of two generational cohorts (N = 24).

For lab strain *Ae. aegypti*, all females oviposited within 1 day. The frequency of females that oviposited at all was dependent on container size (Fig 11A; $\chi^2 = 8.64$, df = 3, $P = 0.034$). There were significant pairwise comparisons where oviposition was dependent on tube size between 1.5-mL and 5-mL tubes ($P = 0.024$), 1.5-mL and 15-mL tubes ($P = 0.024$), and 1.5-mL and 50-mL tubes ($P = 0.005$). No other pairwise comparisons were significant. When oviposition occurred,

**Table 1. Mean±standard error of the mean (SEM) number and percentage of eggs hatched relative to number tested in two strains of *Culex quinquefasciatus* (CMAVE – lab strain; wild – field collected strain) and two strains of *Aedes aegypti* (ORL – lab strain; wild – field collected strain).**

| Species | Strain | Eggs hatched[1] | Eggs tested[1] | % hatched[2] |
|---|---|---|---|---|
| *Cx. quinquefasciatus* | Lab | 14.3±1.29 | 20.3±0.34 | 70.4±6.45 |
| | Wild | 15.3±1.71 | 19.5±0.92 | 75.6±8.12 |
| | Mean±SEM % hatched w/strains pooled=72.0±5.07* | | | |
| Species | Strain | Eggs hatched[1] | Eggs tested[1] | % hatched[2] |
| *Ae. aegypti* | Lab | 18.0±0.70 | 19.7±0.32 | 90.2±3.51 |
| | Wild | 17.6±0.78 | 19.7±0.30 | 89.5±3.59 |
| | Mean±SEM % hatched w/strains pooled=89.9±2.53* | | | |

[1]Units in mean±standard error of the mean (SEM) number of eggs.

[2]Units in mean±standard error of the mean (SEM) % of eggs hatched.

* No statistical difference among tube sizes at α=0.05.

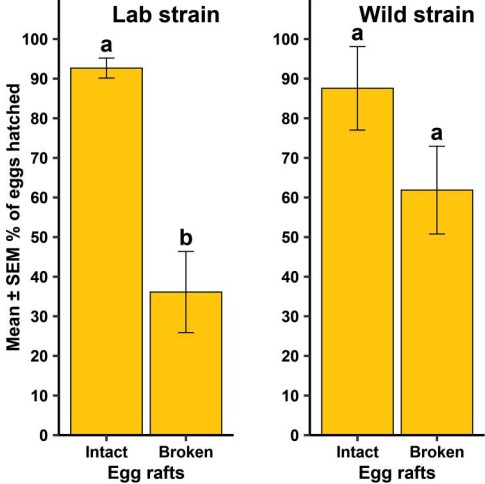

**Fig 6. Mean± standard error of the mean (SEM) % eggs hatched when egg rafts from a lab strain and a wild strain of *Culex quinquefasciatus* were observed broken up versus being a single, intact egg raft.** Different lowercase letters above error bars indicate statistical significance of pairwise comparisons between the egg raft status, with significance set at α=0.05. The lab strain was replicated 13-18 times across each of four generational cohorts (N=60) and the wild strain was replicated 11-13 times across two generational cohorts (N=24).

there was also a significant effect of tube size on the mean number of eggs oviposited (Fig 11B; F=5.21, df=3, 96, P=0.002). Pairwise comparisons of tube size revealed significant differences between 1.5-mL tubes and 15-mL tubes (P=0.002), as well as 1.5-mL and 50-mL tubes (P=0.049). There was no significant difference among the tube sizes in the number of eggs that hatched (Table 2; F=0.10, df=3, 96, P=0.961).

## Discussion

Our study demonstrated that the modified forced oviposition method is a highly effective approach in generating progeny for both *Cx. quinquefasciatus* and *Ae. aegypti* laboratory strains and F$_1$ progeny for field-caught individuals. Forced oviposition has been successfully used to establish wild colonies of multiple *Anopheles spp.* [9–12]. However, this study represents the first report that the technique can be adapted for *Ae. aegypti* and *Cx. quinquefasciatus*. Methods developed

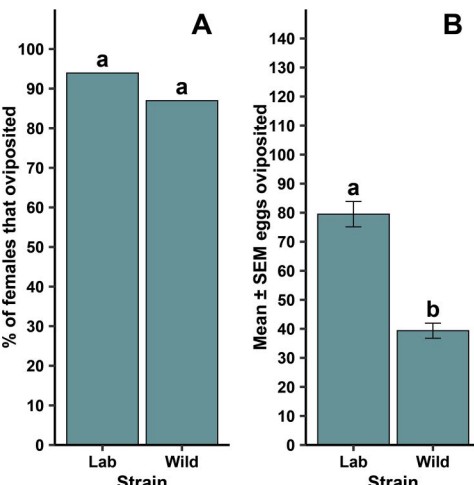

**Fig 7. Differences for forced oviposition success frequency (A) and mean± standard error of the mean (SEM) eggs when lab and wild strain female *Ae. aegypti* oviposited (B).** Different lowercase letters above bars indicate statistical significance of pairwise comparisons between strains, with significance set at α = 0.05. The lab strain was replicated 9 – 13 times across three generational cohorts (N = 33) and the wild strain was replicated 10 – 13 across two generational cohorts (N = 23).

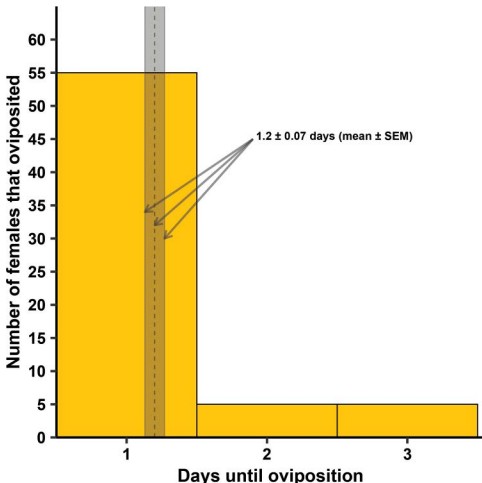

**Fig 8. Number of female lab strain *Cx. quinquefasciatus* that oviposited between 1 and 3 days in Experiment 2, utilizing different tube sizes.** No statistical difference was seen among the tube sizes at α = 0.05, so mean± standard error of the mean (SEM) was calculated across all tube sizes. The mean days to oviposition was 1.2 ± 0.07 (gray column in the figure represents SEM, with the dashed line in the middle representing the mean). All four tube sizes were replicated 8 – 10 times across each of three generational cohorts (N = 114).

for *Culex spp.* and *Ae. aegypti* have been previously reported in the literature but other techniques used larger containers, provided little instruction, required an excessive number of gravid females to ensure colony success, or recommended more specialized equipment, longer setup, and continual upkeep while holding mosquitoes [17,18]. Our results show that the percentage of females that oviposited was higher in *Ae. aegypti* compared to *Cx. quinqufasciatus*, suggesting this method was more effective for *Ae. aegypti*. That being said, > 50% of *Cx. quinquefasciatus* lab and field-caught females successfully oviposited (Fig 4A). Despite differences in performance, this technique, while using a minimal number of

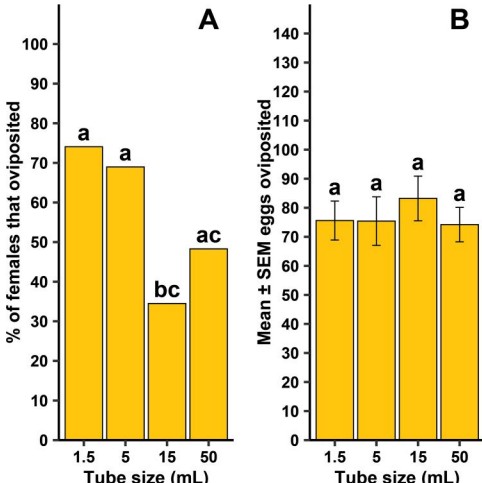

**Fig 9. Tube size effects on forced oviposition success frequency (A) and mean ± standard error of the mean (SEM) eggs when females oviposited (B) by female lab strain *Culex quinquefasciatus*.** Different lowercase letters above error bars indicate statistical significance of pairwise comparisons between different tube sizes, with significance set at α = 0.05. All four tube sizes were replicated 8 – 10 times across each of three generational cohorts (N = 114).

**Table 2. Mean ± standard error of the mean (SEM) number and percentage of eggs hatched relative to number tested in four centrifuge tube sizes for lab strains of *Culex quinquefasciatus* (i.e., CMAVE) and *Aedes aegypti* (i.e., ORL).**

| Species | Tube size (mL) | Eggs hatched[1] | Eggs tested[1] | % hatched[2] |
|---|---|---|---|---|
| *Cx. quinquefasciatus* | 1.5 | 13.6 ± 1.97 | 20.8 ± 0.50 | 64.0 ± 8.96 |
| | 5 | 9.2 ± 2.03 | 21.4 ± 0.93 | 43.6 ± 9.75 |
| | 15 | 12.2 ± 3.04 | 22.2 ± 0.76 | 58.8 ± 14.7 |
| | 50 | 10.7 ± 2.34 | 19.8 ± 1.19 | 66.8 ± 19.4 |
| | Mean ± SEM % hatched w/tube sizes pooled = 57.6 ± 0.06* | | | |
| **Species** | **Tube size (mL)** | **Eggs hatched[1]** | **Eggs tested[1]** | **% hatched[2]** |
| *Ae. aegypti* | 1.5 | 19.1 ± 0.28 | 20.1 ± 0.05 | 95.0 ± 1.38 |
| | 5 | 18.9 ± 0.34 | 20.1 ± 0.09 | 93.8 ± 1.72 |
| | 15 | 18.9 ± 0.43 | 20.1 ± 0.13 | 94.2 ± 2.40 |
| | 50 | 18.6 ± 0.82 | 19.9 ± 0.63 | 92.8 ± 3.45 |
| | Mean ± SEM % hatched w/tube sizes pooled = 94.1 ± 0.01* | | | |

[1]Units in mean ± standard error of the mean (SEM) number of eggs.

[2]Units in mean ± standard error of the mean (SEM) % of eggs hatched.

*No statistical difference among tube sizes at α = 0.05.

female mosquitoes and limited equipment, successfully generated numbers of eggs for both species and strains needed for initial colony establishment and subsequent vector research.

In Experiment 1, both laboratory and field-caught *Cx. quinquefasciatus* females consistently displayed a pattern of longer oviposition times compared to *Ae. aegypti*. Where all *Ae. aegypti* strains oviposited within 24 hrs, *Cx. quinquefasciatus* strains required more time to oviposit, with this pattern dependent on strain displayed by a maximum of 3–7 days to oviposit in laboratory and wild strains, respectively (Fig 4). However, this timeline is not outside the normal observed ranges, as most *Culex spp*. rearing protocols recommend monitoring for egg rafts up to seven days post blood feeding in laboratory colonies [19,20]. In both *Cx. quinquefasciatus* and *Ae. aegypti*, there was no statistical difference in the

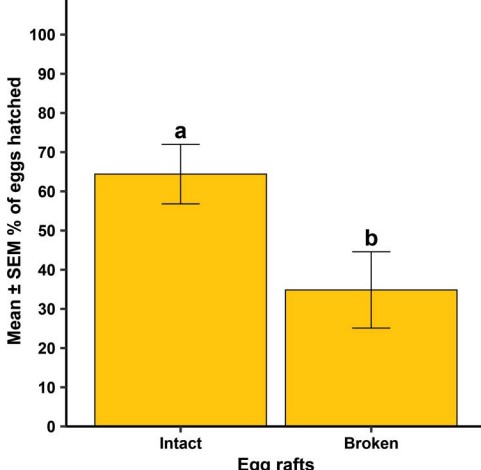

**Fig 10. Mean± standard error of the mean (SEM) % eggs hatched when lab strain *Culex quinquefasciatus* rafts were observed broken up versus being a single, intact egg raft.** Different lowercase letters above error bars indicate statistical significance of pairwise comparisons of percent egg hatching across all tube sizes, with significance set at α = 0.05. All four tube sizes were replicated 8 – 10 times across each of three generational cohorts (N = 114).

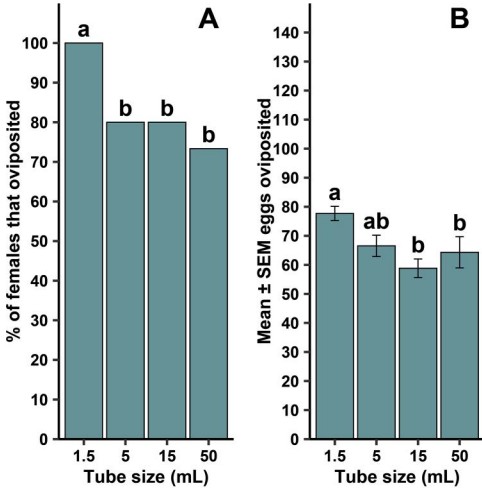

**Fig 11. Tube size effects on forced oviposition success frequency (A) and mean± standard error of the mean (SEM) eggs when females oviposited (B) by gravid female lab strain *Aedes aegypti*.** Different lowercase letters above error bars indicate statistical significance of pairwise comparisons between different tube sizes, with significance set at α = 0.05. All four tube sizes were replicated 15 times across two generational cohorts (N = 120).

percentage of laboratory reared and field-caught females that oviposited. All larvae that hatched were reared to the L2 stage at a growth rate commonly observed in both laboratory and field-caught strains of these species [19,20]. Wild gravid *Cx. quinquefasciatus* females on average laid more eggs or larger egg rafts compared to laboratory strains. Interestingly, the hatch rate of broken egg rafts compared to whole intact egg rafts was significantly reduced in lab strains, while there was no statistical difference in wild *Cx. quinquefasciatus* hatch rate between broken and intact egg rafts. The higher numbers of oviposited eggs and hatch rate of broken egg rafts in the wild strain may be related to the overall robustness of wild strains compared to inbred lab strains. Laboratory environments themselves impose selective pressures on the

population. Standard rearing protocols control all aspects of the mosquito's environment, which, in this case, may lead to a bottleneck effect that selects for whole rafts or increased egg numbers under strictly ideal conditions [21]. The generational exposure to a standardized environment may lead to the selection of characteristics that would be disadvantageous once placed outside their highly regulated laboratory environment [22,23]. On the other hand, in the wild, it may be an adaptive advantage to produce hardier eggs, that are still viable even if the egg raft breaks. Further research needs to be done to investigate the difference in egg viability and hatch rate between intact and broken *Cx. quinquefasciatus* egg rafts.

Tube size did affect oviposition rates, with a pattern of reduced oviposition frequency with an increase in container size. Both species appeared to demonstrate a claustrophilic behavior in oviposition (i.e., displaying an oviposition preference and higher frequency of ovipositing in smaller volume tubes by gravid females), with more females laying in 1.5 and 5 mL tubes for *Cx. quinquefasciatus* and 1.5 mL tubes for *Ae. aegypti*. Additionally, the mean number of eggs laid by *Ae. aegypti* decreased as tube size increased. There was no change in mean egg numbers for *Cx. quinquefasciatus* across all tube sizes. The change in egg numbers across tube size seen in *Ae. aegypti* could be attributed to their 'skip-oviposition' behavior, where females distribute eggs across multiple oviposition sites within a single gonotrophic cycle. This means eggs are laid intermittently and in many cases *Ae. aegypti* females only aggregate a small percentage of their eggs in a single container [24–26]. By comparison, *Cx. quinquefasciatus* typically lay their eggs as a clutch in a single raft, which could be interpreted as an 'all or nothing' approach [27]. To our knowledge, there are no reports of an intermittent style oviposition behavior observed in *Cx. quinquefasciatus*. One limitation in the present study for the claim that mosquitoes exhibit claustrophilic oviposition is that tubes were not selected based on a consistent ratio of increasing size. Instead, tube sizes were selected based on their similar material (i.e, plastic) and common availability in laboratories and mosquito control districts. Thus, with the data generated in the current study, we cannot rule out that tube dimensions may have played a role in oviposition behaviors. Further study, using containers proportionally scaled in size, would strengthen the definition of claustrophilic oviposition.

Based on our data, we recommend the use of 1.5 mL tubes for *Cx. quinquefasciatus* and *Ae. aegypti* egg collection, regardless of the strains being laboratory reared, or field caught. Tube setup should follow the methods outlined in Experiment 1, with the key difference between the two species being the level of excess water left in the tubes (a thin layer of water above the cotton, approx. 1 mm at the meniscus for *Cx. quinquefasciatus*, and complete removal of excess water for *Ae. aegypti*). A key consideration for adopting this method is the quality and type of water that is available. While tap water was used in this study, if a particular water source is deemed unacceptable for rearing purposes, then alternative waters sources may be necessary. Additionally, we recommend leaving *Cx. quinquefasciatus* females to oviposit for 3–4 days for lab strains and up to 7 days for wild strains. *Aedes aegypti* only required 24 hrs post transfer before they oviposited. It is worth noting that the age and degree of blood engorgement of laboratory and field-caught mosquitoes was not controlled for, nor was the longevity of eggs examined in this study. Adult age and level of blood engorgement are key factors that influence oviposition, however, it is difficult to control for these in field caught populations [27]. Further study will need to be conducted to determine if age and blood engorgement significantly impact the success of this forced oviposition method. All *Ae. aegypti* egg papers were hatched within 48 hrs, and *Cx. quinquefasciatus* egg rafts were not held in oviposition tubes past 24 hrs. For the purposes of this paper, we recommend readers adhere to the same timeline for egg hatching, i.e., *Ae. aegypti* papers dried for 48 hrs before hatching and *Cx. quinquefasciatus* egg rafts placed in water within 24 hrs.

The present study demonstrates that the forced oviposition method can be adapted to other mosquito vector species besides *Anopheles spp*. Present challenges in mosquito vector research start initially with ensuring teams have access to sufficient numbers of mosquitoes and successful production of $F_1$ progeny required for colony establishment of field strains. This technique was shown to be highly effective in generating viable eggs using a minimal number of female mosquitoes and readily available items found in most laboratories and mosquito control districts. Proper utilization of the forced oviposition method expands our ability to conduct a variety of vector research with multiple key mosquito vector species, such as single female sex ratio determination, genetic isoline production by controlling for specific genetic backgrounds, and heritability studies focused on selective mating.

## Acknowledgments

We thank Lyle Buss and Randy Fernandez for assisting in photography and graphic design of images used in Figs 1–3. We thank Neil Sanscrainte for technical assistance in the laboratory.

## Author contributions

**Conceptualization:** Kendra A. Dagg, Edwin Burgess.

**Data curation:** Kendra A. Dagg.

**Formal analysis:** Edwin Burgess.

**Funding acquisition:** Edwin Burgess.

**Investigation:** Kendra A. Dagg.

**Methodology:** Kendra A. Dagg, Alden S. Estep, Cason E. Bartz, Edwin Burgess.

**Project administration:** Edwin Burgess.

**Resources:** Alden S. Estep, Cason E. Bartz, Edwin Burgess.

**Software:** Edwin Burgess.

**Supervision:** Edwin Burgess.

**Validation:** Kendra A. Dagg, Edwin Burgess.

**Visualization:** Kendra A. Dagg, Edwin Burgess.

**Writing – original draft:** Kendra A. Dagg, Edwin Burgess.

**Writing – review & editing:** Kendra A. Dagg, Alden S. Estep, Cason E. Bartz, Edwin Burgess.

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
