## [Decision Letter · Decision Letter 0]

Claustrophilic oviposition: oviposition performance depends on container size in a novel forced oviposition method for Culex quinquefasciatus and Aedes aegypti

Dear Dr. Burgess,

Thank you for submitting your manuscript to PLOS Neglected Tropical Diseases. After careful consideration, we feel that it has merit but does not fully meet PLOS Neglected Tropical Diseases's publication criteria as it currently stands. Therefore, we invite you to submit a revised version of the manuscript that addresses the points raised during the review process.

Please submit your revised manuscript within 60 days Jul 12 2025 11:59PM. If you will need more time than this to complete your revisions, please reply to this message or contact the journal office at plosntds@plos.org. Please include the following items when submitting your revised manuscript:

We look forward to receiving your revised manuscript.

Kind regards,

Adly M.M. Abd-Alla, Prof asso.

Section Editor

Adly Abd-Alla

Section Editor

Shaden Kamhawi

co-Editor-in-Chief

Paul Brindley

co-Editor-in-Chief

**Journal Requirements:**

At this stage, the following Authors/Authors require contributions: Kendra A. Dagg, Alden S. Estep, Cason E. Bartz, and Edwin Burgess. Please ensure that the full contributions of each author are acknowledged in the "Add/Edit/Remove Authors" section of our submission form.

- ® on page: 7.

Potential Copyright Issues:

- Please confirm (a) that you are the photographer of Figures 1, 2, and 3, or (b) provide written permission from the photographer to publish the photo(s) under our CC BY 4.0 license.

5) Please amend your detailed Financial Disclosure statement. This is published with the article. It must therefore be completed in full sentences and contain the exact wording you wish to be published. Please ensure that the funders and grant numbers match between the Financial Disclosure field and the Funding Information tab in your submission form. Note that the funders must be provided in the same order in both places as well.

**Reviewers' Comments:**

Reviewer's Responses to Questions

**Key Review Criteria Required for Acceptance?**

**Methods:**

-Are the objectives of the study clearly articulated with a clear testable hypothesis stated?

-Is the study design appropriate to address the stated objectives?

-Is the population clearly described and appropriate for the hypothesis being tested?

-Is the sample size sufficient to ensure adequate power to address the hypothesis being tested?

-Were correct statistical analysis used to support conclusions?

-Are there concerns about ethical or regulatory requirements being met?

Reviewer #1: 1. The study’s aim of “comparing egg numbers” is not associated with a clearly defined threshold; it fails to distinguish expected outcomes for oviposition rate, egg number, and hatch rate. Additionally, potential confounding factors such as mosquito age or degree of blood engorgement were not controlled for in assessing oviposition performance.

2. The definition and observation criteria for "claustrophilic" behavior lack rigorous quantification.

3. The tube sizes used in Experiment 2 (1.5, 5, 15, and 50 mL) are not proportionally scaled or logically progressive, making it difficult to establish a consistent gradient for interpretation. Furthermore, moisture conditions differed between species—Cx. tubes retained a surface water layer, whereas Ae. tubes had excess water removed—which could significantly affect oviposition behavior. The criteria for determining oviposition were not clearly described, including whether this was assessed visually and whether all germination papers were collected and systematically examined.

Reviewer #2: Yes the methods are well described and appropriate

Reviewer #3: The nature of the wild mosquitos need more description (time collected, location, identification method...). Was the tap water used dechlorinated, fluoridated, harvested rain water or borehole water?

**Results**

-Does the analysis presented match the analysis plan?

-Are the results clearly and completely presented?

-Are the figures (Tables, Images) of sufficient quality for clarity?

Reviewer #1: (No Response)

Reviewer #2: Yes to the above

Reviewer #3: Well presented and of sufficient quality.

**Conclusions:**

-Are the conclusions supported by the data presented?

-Are the limitations of analysis clearly described?

-Do the authors discuss how these data can be helpful to advance our understanding of the topic under study?

-Is public health relevance addressed?

Reviewer #1: The conclusion that the mosquitoes “exhibited claustrophilic behavior” is not well-supported, as it is based solely on oviposition rates without behavioral recordings or exploratory data. The study also lacks discussion of potential effects of the method on mosquito longevity or offspring quality, and does not address possible impacts on sex ratio or population fitness in downstream research.

Reviewer #2: yes to the above

Reviewer #3: (No Response)

**Editorial and Data Presentation Modifications?**

Reviewer #1: (No Response)

Reviewer #2: (No Response)

Reviewer #3: (No Response)

**Summary and General Comments:**

Reviewer #1: This study proposed and evaluated a modified forced oviposition method applicable to Culex quinquefasciatus and Aedes aegypti, finding that both species exhibited higher oviposition success rates in small-volume containers and displayed "claustrophilic" oviposition behavior. The method uses inexpensive materials and a single-mosquito setup, making it suitable for oviposition induction in both laboratory and wild mosquito populations, and holds promise for facilitating colony establishment and genetic experiments in vector research.

The background lacks a systematic review of existing forced oviposition methods, their limitations, and comparative analyses. The "claustrophilic" hypothesis was introduced late in the manuscript instead of being explicitly stated and biologically contextualized in the introduction. There is no literature-based or physiological support for the mechanism of “small space–induced oviposition,” making the hypothesis largely speculative.

Reviewer #2: This is a very well written detailed paper that can be useful for programs that has important implications for enahcing vector-borne disease research. Minor edits are:

Line 108 and 109 in Materials and Methods: Culex and Aedes already mentioned so can abbreviate the genus

Line 290 aegypti is capitalized

Could explain in line 307 what claustrophilic behavior is

Reviewer #3: The study presents a practical forced oviposition method adapted for Culex and Aedes species adapted from Anopheles methods highlighting its usefulness in generating sufficient F1 eggs for colony establishment despite their differing natural oviposition behaviors.

It is well written and easy for the reader to follow through and replicate the experiments.

The technique’s simplicity—requiring minimal female mosquitoes and low-cost lab materials—makes it highly accessible for vector control programs and various research settings.

A more elaborate description of the wild strains used will enhance repeatability and reveal any confounding factors on the outcome of the experiments including habitat type and environmental conditions pre-testing that might affect behavioral traits.

PLOS authors have the option to publish the peer review history of their article (what does this mean? ). If published, this will include your full peer review and any attached files.

**Do you want your identity to be public for this peer review?** For information about this choice, including consent withdrawal, please see our Privacy Policy .

Reviewer #1: No

Reviewer #2: No

Reviewer #3: No

**Figure resubmission:**

**Reproducibility:**



---

## [Decision Letter · Decision Letter 1]

Dear Dr. Burgess,

We are pleased to inform you that your manuscript 'Claustrophilic oviposition: oviposition performance depends on container size in a novel forced oviposition method for Culex quinquefasciatus and Aedes aegypti' has been provisionally accepted for publication in PLOS Neglected Tropical Diseases.

Best regards,

Adly M.M. Abd-Alla, Prof asso.

Section Editor

Adly Abd-Alla

Section Editor

Shaden Kamhawi

co-Editor-in-Chief

Paul Brindley

co-Editor-in-Chief

Reviewer's Responses to Questions

**Key Review Criteria Required for Acceptance?**

**Methods**

-Are the objectives of the study clearly articulated with a clear testable hypothesis stated?

-Is the study design appropriate to address the stated objectives?

-Is the population clearly described and appropriate for the hypothesis being tested?

-Is the sample size sufficient to ensure adequate power to address the hypothesis being tested?

-Were correct statistical analysis used to support conclusions?

-Are there concerns about ethical or regulatory requirements being met?

Reviewer #1: (No Response)

Reviewer #2: The methods are clear and the minor edits have made improvements to the manuscript.

Reviewer #3: (No Response)

**Results**

-Does the analysis presented match the analysis plan?

-Are the results clearly and completely presented?

-Are the figures (Tables, Images) of sufficient quality for clarity?

Reviewer #1: (No Response)

Reviewer #2: The results are clear.

Reviewer #3: (No Response)

**Conclusions**

-Are the conclusions supported by the data presented?

-Are the limitations of analysis clearly described?

-Do the authors discuss how these data can be helpful to advance our understanding of the topic under study?

-Is public health relevance addressed?

Reviewer #1: (No Response)

Reviewer #2: Improvements to the conclusions have been made and increased the clarity of the findings.

Reviewer #3: (No Response)

**Editorial and Data Presentation Modifications?**

Reviewer #1: (No Response)

Reviewer #2: (No Response)

Reviewer #3: (No Response)

**Summary and General Comments**

Reviewer #1: (No Response)

Reviewer #2: (No Response)

Reviewer #3: The review concerns have been sufficiently addressed and this is a much improved version of the MS.

PLOS authors have the option to publish the peer review history of their article (what does this mean? ). If published, this will include your full peer review and any attached files.

**Do you want your identity to be public for this peer review?** For information about this choice, including consent withdrawal, please see our Privacy Policy .

Reviewer #1: No

Reviewer #2: No

Reviewer #3: No

---

## [Editor Report · Acceptance letter]

Dear Dr. Burgess,

We are delighted to inform you that your manuscript, "Claustrophilic oviposition: oviposition performance depends on container size in a novel forced oviposition method for Culex quinquefasciatus and Aedes aegypti," has been formally accepted for publication in PLOS Neglected Tropical Diseases.

Best regards,

Shaden Kamhawi

co-Editor-in-Chief

Paul Brindley

co-Editor-in-Chief
